# Association between the *ABCC11* gene polymorphism-determined earwax properties and external auditory canal microbiota in healthy adults

Yasunobu Amari,[1,2] Masahiro Hosonuma,[1,3,4,5] Takuya Mizukami,[1] Junya Isobe,[6] Yuki Azetsu,[5,7] Eiji Funayama,[3,5,8] Yuki Maruyama,[1,3,5] Toshiaki Tsurui,[1,3,4,5] Kohei Tajima,[3,9] Aya Sasaki,[1,3,5] Yoshitaka Yamazaki,[5,8,10] Ryota Nakano,[11] Yutaka Sano,[12] Atsushi Ishida,[1,5,13] Tatsuya Nakanishi,[1,5,13] Seiji Mochizuki,[1] Yuri Yoshizawa,[1] Sumito Kumagai,[1] Sakiko Yasuhara,[1] Kakei Ryu,[1] Tatsunori Oguchi,[1,5] Atsuo Kuramasu,[3] Kiyoshi Yoshimura,[3,4] Takehiko Sambe,[1,14] Sei Kobayashi,[2] Naoki Uchida[1]

**ABSTRACT** The concept of genome–microbiome interactions, in which the microenvironment determined by host genetic polymorphisms regulates the local microbiota, is important in the pathogenesis of human disease. In otolaryngology, the resident bacterial microbiota is reportedly altered in non-infectious ear diseases, such as otitis media pearls and exudative otitis media. We hypothesized that a single-nucleotide polymorphism in the ATP-binding cassette sub-family C member 11 (*ABCC11*) gene, which determines earwax properties, regulates the ear canal microbiota. We analyzed *ABCC11* gene polymorphisms and ear canal microbiota in healthy individuals to understand the relationship between genome–microbiome interactions in the ear canal. The study included 21 subjects who were divided into two groups: 538GA (9) and 538AA (12). *Staphylococcus auricularis* and *Corynebacterium* spp. were observed in the 538GA group, whereas *Methylocella* spp. was observed in the 538AA group. PICRUSt analysis revealed significant enrichment of certain pathways, such as superpathway of N-acetylglucosamine, N-acetylmannosamine and N-acetylneuraminate degradation, chlorosalicylate degradation, mycothiol biosynthesis, and enterobactin biosynthesis in the GA group, whereas allantoin degradation IV (anaerobic), nitrifier denitrification, starch degradation III, L-valine degradation I, and nicotinate degradation I were significantly enriched in the AA group. The *ABCC11* gene polymorphism regulates the composition of the ear canal microbiota and its metabolic pathways. This study revealed a genome–microbiome interaction within the resident microbiota of the external auditory canal that may help to elucidate the pathogenesis of ear diseases and develop novel therapies.

**IMPORTANCE** The *ABCC11* gene polymorphism, which determines earwax characteristics, regulates the composition of the ear canal microbiota and its metabolic pathways. We determined the presence of genome–microbiome interactions in the resident microbiota of the ear canal. Future studies should focus on *ABCC11* gene polymorphisms to elucidate the pathogenesis of ear diseases and develop therapeutic methods.

**KEYWORDS** *ABCC11*, human microbiome, genome-microbiome interaction, ear canal

Bacteria have co-existed and co-evolved with eukaryotes for 3.5 billion years. In humans, commensal bacteria, observed primarily in the nasal cavity, oral cavity, gastrointestinal tract, and skin affect the host through their components and metabolites. Studies on human intestinal bacteria have revealed their significant role in determining the risk of disease onset and treatment response (1–4). In contrast, alterations in resident bacterial microbiota have been reported in non-infectious ear

**Peer Reviewer** Funmilola Abidemi Ayeni, Indiana University Bloomington, Bloomington, Indiana, USA

Address correspondence to Masahiro Hosonuma, masa-hero@med.showa-u.ac.jp.

The authors declare no conflict of interest.

See the funding table on p. 8.

diseases. *Alternaria* spp. and *Cladosporium herbarum* are less abundant in the middle ear of patients with otitis media pearls (5). Additionally, *Alloiococcus otitidis* and *Haemophilus* are common in the middle ear of patients with exudative otitis media (6, 7). However, factors that determine the resident bacterial microbiota in the middle ear cavity and ear canal remain unexplored.

Host genetic polymorphisms determine endogenous bacteria, resulting in genome–microbiome interactions (8–10). This concept is crucial for understanding the symbiosis between humans and bacteria because, unlike environmental factors, host genes are conserved throughout life. In humans, studies have identified resident microbiome-associated genetic polymorphisms at 10 sites, including stool, anterior nares, hard palate, palatine tonsils, saliva, supragingival plaque, throat, and tongue dorsum (8). Additionally, in disease pathogenesis, the nucleotide-binding oligomerization domain-containing protein 2 (*NOD2*) gene, a risk allele for inflammatory bowel disease, is associated with the concentration of *Lactobacillus* and *Bacteroides uniformis* in the gut, whereas in primary sclerosing cholangitis, the fucosyltransferase 2 (*FUT2*) gene is associated with the concentration of *Firmicutes* and *Proteobacteria* in the bile (11, 12). This indicates that elucidation of organ-specific genome–microbiome interactions is important for understanding disease pathogenesis. However, to our knowledge, there are no reports on genome–microbiome interactions in the resident microbiota of the external auditory canal.

The composition of commensal bacteria is determined by the various substrates present in their microenvironments. In the intestinal tract, diet determines the composition of intestinal bacteria because they metabolize components of digestive residues in the colon as substrates (13). In contrast, we hypothesized that variations in the properties of earwax (a potential substrate) may alter the composition of bacterial microbiota of the external auditory canal. The human ear canal contains wet and dry earwax, which is formed when secretions (lipids and peptides) from sebaceous and apocrine sweat glands mix with keratinized epidermal cells that are expelled by the self-cleaning action of the ear canal. The ratio of these glands is determined by a single-nucleotide polymorphism (SNP) (538G > A) in the ATP-binding cassette subfamily C member 11 (*ABCC11*) gene on chromosome 16 (14–17), which determines the nature of earwax—wet (538GG and 538GA) and dry (538AA) (18). In this study, we simultaneously analyzed the *ABCC11* gene polymorphisms and bacterial microbiota of the ear canal in healthy individuals to understand the relationship between earwax characteristics and ear canal microbiota as genome–microbiome interactions in the ear canal.

## MATERIALS AND METHODS

### Selection of subjects

Twenty-one healthy adults working at Showa University from April to June 2023, aged 20–50 years, were included in this study. Individuals with ear disease complications, malignant disease, or treatment with antimicrobials or steroids within 1 month were excluded. The characteristics of earwax were defined by self-declaration, as reported previously (18).

### Data collection

The following clinical information was collected: age, sex, underlying disease, medication history (antimicrobials and steroids), presence of ear disease, ear-cleaning habits, and self-perceived earwax characteristics. Ear cleaning habits were categorized as low frequency (up to twice a week) and high frequency (three or more times a week).

### Genetic analysis

Genomic DNA was extracted from cells collected from the oral buccal mucosa, stored in a phosphate buffer solution, and subsequently dissolved in ISOGENOME (code no. 314–

08113; NIPPON GENE). PCR was performed using KOD FX (code no. KFX-101; TOYOBO) following the manufacturer's instructions. SNP analysis used specified primers (forward: T TCAGTGCTTCTGGTGATGC and reverse: CCACCATGCTCTCTACTGTCCTC). The PCR products were purified using a High Pure PCR Product Purification Kit (catalog no. 11732668001; Sigma-Aldrich, St. Louis, Missouri, United States). DNA sequence analysis was performed by FASMAC, Inc. The results were assessed by confirming the nucleotide sequence at position 538 on chromosome 16 using a plasmid editor (ApE) (Jorgensen Laboratories, Loveland, Colorado, United States).

## Bacterial microbiota analysis

Samples were collected by gently swabbing the left external auditory canal with a cotton swab for at least three rotations. Subsequently, the samples were stored in a sterile container containing guanidine hydrochloride solution and frozen at −30°C. DNA was extracted using the QIAamp PowerFecal Pro DNA Kit (QIAGEN, Hilden, Germany), following the manufacturer's instructions. Genomic analysis was conducted using a next-generation sequencer (MySeq: Illumina, San Diego, CA, USA) to sequence the V3 and V4 regions of 16SrRNA genes. For bacterial microbiota analysis and functional prediction, sequence data in the FASTQ format were imported into QIIME2 (https://qiime2.org/). Exploratory statistical analysis assessed variations in bacterial abundance between the groups. The R package, QIIME2 (https://github.com/jbisanz/qiime2R) was used for the bacterial community analysis. Functional prediction was performed using PICRUSt2, an extension of QIIME2, to generate the MetaCyc metabolic pathway. Statistical analysis was performed using the Mann–Whitney U test with R version 4.0.5 (https://www.r-project.org/). PCoA establishment ellipses were analyzed with 95% confidence intervals using R. The sequence data for the external auditory canal bacteria gene are available in the DDJB database (https://www.ddbj.nig.ac.jp/index.html) under accession number PRJDB18837.

## Statistical analysis

Statistical analysis was conducted using JMP Pro V.17.0. Categorical data are expressed as numbers and percentages. Kruskal–Wallis test was used to compare *ABCC11* polymorphisms and sex with the proportions of each microbiota. Simple linear regression model was used to compare age at sample collection and proportion of each microbiota. Pearson correlation coefficients were calculated to assess relationships among variables. One-way analysis of variance (ANOVA) with Tukey–Kramer's honestly significant difference (HSD) test was used to compare ear-cleaning habits and the proportions of each microbiota. Statistical significance was set at $P < 0.05$.

## RESULTS

Bacterial microbiota analysis was conducted on samples from all 21 participants (Fig. S1). *ABCC11* gene polymorphisms were observed in the 538GA (9) and 538AA (12) groups. The previously reported relationship between earwax characteristics and *ABCC11* gene polymorphisms was consistent across all cases (Table 1). Initially, principal component analysis was conducted to compare the β-diversity of the 21 samples. Additionally, we compared the ear canal microbiota diversity between the GA and AA groups. Three-dimensional (3D) analysis of β-diversity demonstrated that various clusters were formed between the ear canal microbiota of the GA and AA groups (Fig. 1; Fig. S2). Subsequently, we analyzed the composition of the ear canal microbiota in the GA and AA groups at the species level. The results demonstrated the bacterial species composition of the ear canal in each sample and group and revealed various compositional proportions of certain bacterial species between the GA and AA groups (Fig. 2a and b). In the *ABCC11* gene polymorphisms, *S. auricularis* and *Corynebacterium* spp. were observed in the GA group, whereas *Methylocella* spp. and uncultured fungus were observed in the AA group (Fig. 3a and b; Fig. S3). In the PICRUSt analysis, certain pathways, such as superpathway

**TABLE 1** Basic demographic and clinical details of the study participants

|  | Total ($n = 21$) | AA ($n = 12$) | GA ($n = 9$) |
|---|---|---|---|
| Age (mean ± SD) | 34.1 ± 6.2 | 33.2 ± 6.9 | 35.4 ± 5.3 |
| Female (%) | 38.1 | 33.3 | 44.4 |
| Ear cleaning habits |  |  |  |
| None | 2 | 2 | 0 |
| Low frequency | 13 | 7 | 6 |
| High frequency | 6 | 3 | 3 |

of N-acetylglucosamine, N-acetylmannosamine and N-acetylneuraminate degradation, chlorosalicylate degradation, mycothiol biosynthesis, and enterobactin biosynthesis, were significantly enriched in the GA group, whereas allantoin degradation IV (anaerobic), nitrifier denitrification, starch degradation III, L-valine degradation I, and nicotinate degradation I were significantly enriched in the AA group (Table 2) (Fig. S4). These results indicate significant variations in the composition of the ear canal bacterial microbiota based on the *ABCC11* gene polymorphism-determined earwax characteristics and overall metabolic pathways.

The study included 13 males and 8 females, aged 25–46 years (mean: 34.1 years, median: 33 years) (Table 1). *Streptococcus* spp. were significantly higher in females (*P* = 0.0231), whereas no bacteria were observed at significant levels in males (Fig. 4a). Additionally, the proportion of *Streptococcus* spp. increased with age ($R^2$ = 0.2703, *P* = 0.0157; Fig. 4b). The proportion of *Malassezia* spp. and uncultured fungus were significantly higher in the non-ear-cleaning habit group than that in the low- and high-frequency groups (Fig. 4c; Fig. S3b). The composition of the ear canal microbiota was related to sex and age, indicating that the ear canal fungal microbiota is dependent on ear-cleaning habits.

## DISCUSSION

We observed that the *ABCC11* gene polymorphism, which determines earwax characteristics regulates the composition of the ear canal microbiota and its metabolic pathways in healthy individuals. In addition, there were unique commensal bacteria present only in each *ABCC11* gene polymorphism. The composition of commensal bacteria is determined by the various substrates present in their microenvironments. It has been reported that earwax present in the external auditory canal contains a wide variety of peptides, supporting our results that the different properties of earwax as substrates altered the composition of the bacterial microflora of the external auditory canal (19).

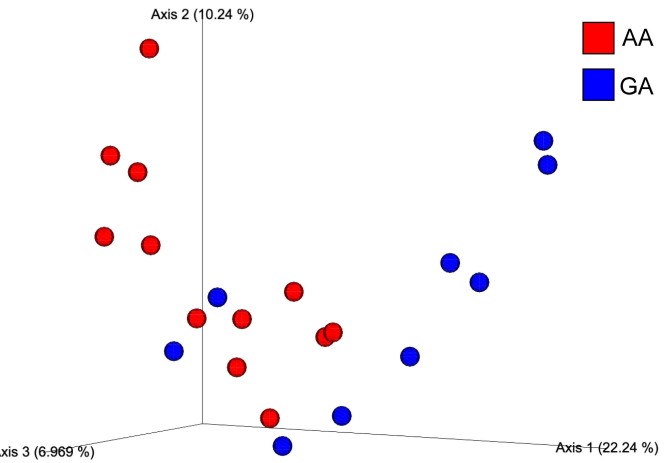

**FIG 1** Association of *ABCC11* gene polymorphisms with the bacterial microbiota of the external auditory canal beta diversity in the ear canal bacterial microbiota of healthy subjects (nine in the AA group and 12 in the GA group) analyzed using QIIME2.

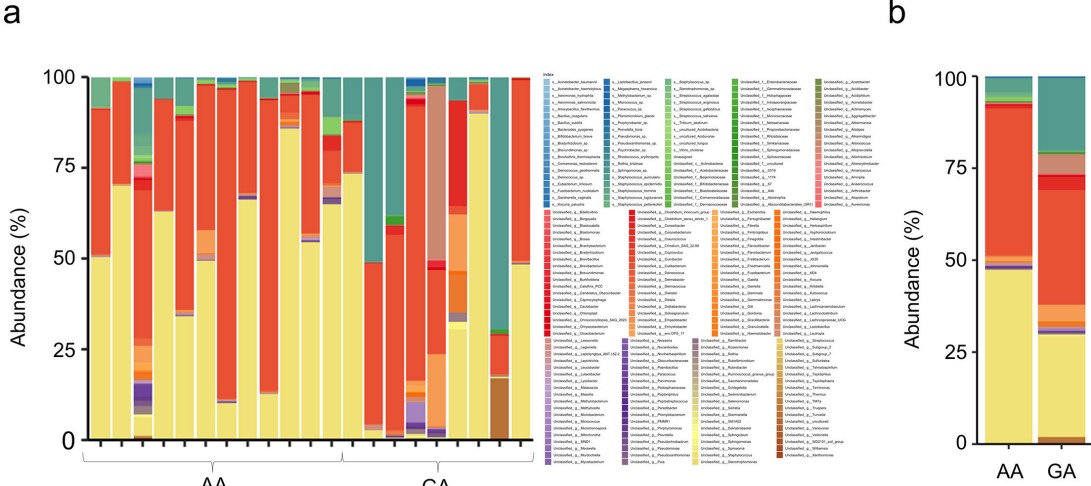

**FIG 2** Relative abundance of ear canal bacteria in each subject. (a). Proportion of bacteria at the discernible genus level in the ear canal of each subject. Names of the bacteria are presented in the bar graph. (b). Bar graph demonstrating the microbiota composition in each group, where the sum of all the bacteria observed in more than 0.1% of the cases in each group is summed to 100% (Fig. 2).

The microbiota of the external auditory canal has been reported in healthy Caucasians, with *S. auricularis* identified as the predominant bacterium in every report (20–22). Since 95% of Caucasians had wet cerumen, our results in which *S. auricularis* was identified only in the GA group were consistent with these previous reports. These results suggest that genetic polymorphisms regulate the bacterial flora across racial groups. The importance of genome–microbiome interactions has been reported in human intestinal and biliary tract diseases (11, 12). On the other hand, one of the ear diseases, otitis media pearls, a benign tumor with bone destruction of the middle ear, originated from the outer ear epithelium, suggesting that the environment of the ear canal plays an important role in its pathogenesis. It has been reported that the *ABCC11* gene polymorphisms 538GG and 538GA are risk factors for middle ear cholesteatoma (23). Furthermore, *Alternaria* spp. and *C. herbarum* were less abundant microbiota of the external auditory canal of subjects with middle ear cholesteatoma compared with healthy subjects (5). Our results may help in understanding the causal relationship between these independent reports on gene polymorphisms and microbiome for middle ear cholesteatoma. In other words, alterations in the ear canal microbiota owing to *ABCC11* gene polymorphisms may alter the microenvironment along with its metabolites, thereby potentially contributing to the development of otitis media pearls as a mechanism in the pathogenesis of cholesteatoma. In the future, simultaneous

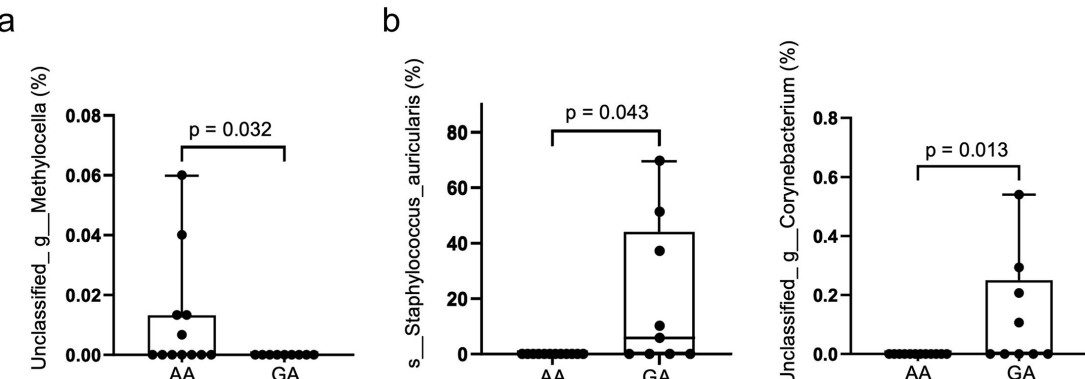

**FIG 3** Differences in the indigenous bacteria of the ear canal owing to *ABCC11* gene polymorphisms (a). Comparison of the relative abundance of *Methylocella* spp. in the AA and GA groups. (b). Comparison of the relative abundance of *Staphylococcus auricularis* and *Corynebacterium* spp. in the AA and GA groups.

**TABLE 2** Differences in metabolic pathways between GA and AA groups based on PICRUSt analysis

| Pathway | Log2 fold change | *P*-value |
|---|---|---|
| Allantoin degradation IV (anaerobic) | −22.9 | <0.001 |
| Superpathway of N-acetylglucosamine, N-acetylmannosamine and N-acetylneuraminate degradation | 2.15 | 0.017 |
| Chlorosalicylate degradation | 2.04 | 0.018 |
| Nitrifier denitrification | −5.87 | 0.020 |
| Enterobactin biosynthesis | 2.14 | 0.037 |
| Mycothiol biosynthesis | 1.82 | 0.037 |
| Starch degradation III | −5.74 | 0.043 |
| L-valine degradation I | −3.72 | 0.047 |
| Nicotinate degradation I | −3.11 | 0.048 |

examination of genetic polymorphisms and ear canal microbiome in ear disease is expected to help elucidate pathogenesis based on genome–microbiome interactions.

In this study, *S. auricularis* and *Corynebacterium* spp. were observed only in the GA group. These two bacteria and the *ABCC11* polymorphism have been reported to be associated with axillary osmidrosis. The axillary osmidrosis originates from the presence of odorant E3M2H in apocrine sweat as a precursor (nonvolatile glutamine conjugate), which is secreted into the axillary skin. Subsequently, the amino acid bonds are cleaved by *Corynebacterium* spp.-produced Nα-acylglutamine aminoacylase to form volatile compounds responsible for the odor (24, 25). In the axillary skin commensal flora of axillary osmidrosis, a higher abundance of *Staphylococcus* spp. was found in female patients, and a higher abundance of *Corynebacterium* spp. was found in male patients compared with healthy subjects (26). Additionally, axillary osmidrosis was strongly associated with the wet earwax genotype (GA/GG) due to its involvement in apocrine gland secretory cell function (27). These reports and our results indicate that the *ABCC11* gene polymorphism may have a common effect on the indigenous microbiota of the axillary skin and external auditory canal, where apocrine glands are located, and may contribute to the risk of disease development. On the other hand, *Corynebacterium otitidis* has been reported as one of the causative organisms of acute otitis media (21, 28–31). Infants of Caucasian descent have been reported to have a higher incidence of acute

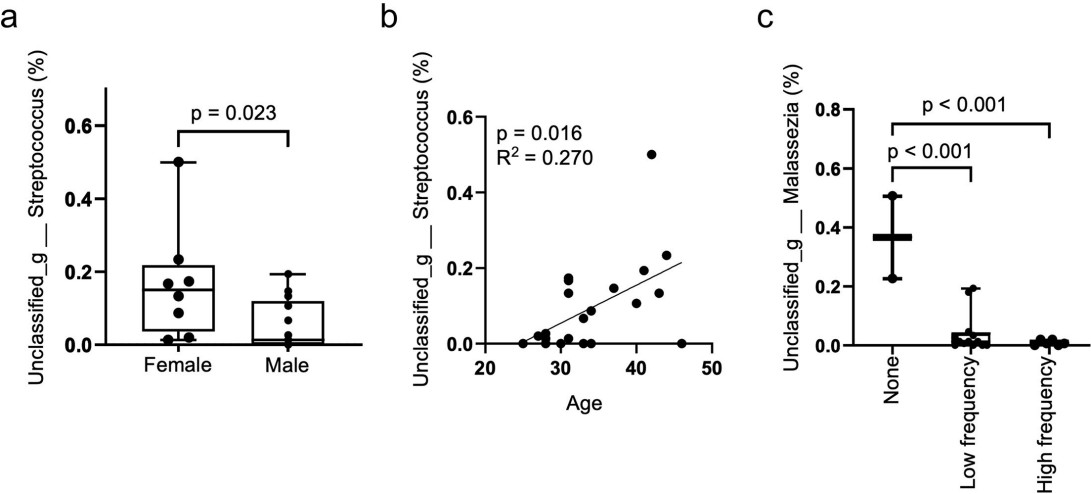

**FIG 4** Differences in resident bacteria in the ear canal by sex, age, and ear-cleaning habits (a). Comparison of the relative abundance of *Streptococcus* spp. by sex. (b). Comparison of the relative abundance of *Streptococcus* spp. by age. Simple linear regression model was used to compare age at sample collection and proportion of each microbiota. Pearson correlation coefficients were calculated to assess relationships among variables. (c). Comparison of the relative abundance of *Malassezia* spp. by ear-cleaning habits. One-way analysis of variance (ANOVA) with Tukey–Kramer's HSD test was used to compare ear-cleaning habits and the proportions of each microbiota.

otitis media than infants of Asian descent (32). Since the majority of Caucasians have wet earwax, we speculate that *Corynebacterium* sp. identified only in the GA group in this study may have contributed to the increased incidence as the causative organism. These findings indicate that polymorphisms in the *ABCC11* gene may explain the variation in the incidence of acute otitis media among racial groups.

In this study, *Methylocella* spp. and uncultured fungus were observed only in the AA group. *Methylocella* spp. is a typical methanotrophic bacterium, but its symbiosis with mammals and its effects on the host have not been reported. Future studies are needed to determine whether the microenvironment of the ear canal, with its dry earwax, is suitable for *Methylocella* spp. to colonize, or whether they also act on human cells and establish a symbiotic relationship. The fungal data in this study are limited to fungus with 16S ribosomal RNA, so it is not possible to discuss most of the fungus. In recent years, research on the gut mycobiome and disease pathology in humans has been focused on (33). In the future, ITS rRNA sequencing should be used to clarify the fungal flora of the external auditory canal.

In this study, several metabolic pathways of the ear canal flora were enriched depending on the *ABCC11* gene polymorphism. Previous reports on gut bacteria and metabolic pathways associated with stool consistency in patients with rectal cancer have found loose/liquid stools may be linked to mycothiol biosynthesis pathways and to *Staphylococcus* abundance (34). This pathway enrichment and increase in *Staphylococcus* are common to the results in the GA group of our study and suggest that the microenvironment with high water content may have affected the function and composition of the bacterial flora both in the intestinal tract and the external auditory canal. The GA group was also enriched in the enterobactin biosynthesis pathway; it has been reported that enterobactin induces the chemokine, interleukin-8, from intestinal epithelia by chelating intracellular iron (35, 36). On the other hand, the L-valine degradation I pathway, which was enriched in the AA group, promotes isobutyric acid production. We previously reported that isobutyric acid induces activation of human T cells in the context of cancer immunity (37). It is suggested that isobutyric acid may act on host cells with its receptors in the external auditory canal. Thus, it is suggested that metabolic changes in the entire flora of the external auditory canal may act on the cells of the external auditory canal *via* its metabolites.

Several background factors were associated with ear canal bacterial flora in this study. In this study, the abundance of *Streptococcus* spp. was significantly higher in females. Various ear diseases, such as otosclerosis, ear canal cancer, and age-related hearing loss are reportedly associated with sex differences, and estrogen has been implicated in some of these diseases (38–42). In contrast, the composition of the intestinal microbiota is altered post-menopause owing to estrogen deficiency (43). Therefore, variations in the bacterial microbiota of the ear canal through sex-dependent differences in estrogen expression may be involved in the development of ear diseases with prevalent sex differences. The percentage of *Streptococcus* spp. in the external auditory canal in this study was positively correlated with age. It has been reported that the intestinal microbiota changes with age and increases cardiovascular disease risk (44, 45). It was suggested that age-related changes in the ear canal microbiota may be related to the pathogenesis of ear diseases that develop in old age. The ear cleaning group had fewer fungi than the group that was not cleaned. Frequent ear cleaning has been reported to increase the risk of otomycosis (46). It is speculated that frequent ear cleaning may increase the risk of otomycosis by reducing resident fungi and facilitating the colonization of pathogenic fungi.

## Conclusions

The *ABCC11* gene polymorphism, which determines earwax properties, may influence the composition of the ear canal microbiota and its metabolic pathways. This study indicates the existence of genome–microbiome interactions in the resident microbiota of the external auditory canal. Future studies are anticipated to focus on the *ABCC11* gene

polymorphism to elucidate the pathogenesis of ear diseases and develop therapeutic methods.

## ACKNOWLEDGMENTS

This study was supported by the Showa University Research Grant for Young Researchers to Y. Amari from Showa University.

Y.A. and M.H. contributed to the conception and design; T.M., J.I., Y.A., E.F., Y.M., T.T., K.T., Y.Y., R.N., Y.S., A.I., T.N., S.M., Y.Y., S.K., S.Y., K.R., T.O., A.K., K.Y., T.S., S.K., and N.U. contributed to data analysis and interpretation. Y.A. contributed to healthy subject recruitment, sample collection, and data acquisition. Y.A. and M.H. drafted the manuscript. All authors have read the journal's authorship agreement and the manuscript has been reviewed by and approved by all named authors.

## AUTHOR AFFILIATIONS

[1]Department of Pharmacology, Showa University Graduate School of Medicine, Shinagawa, Tokyo, Japan
[2]Department of Otolaryngology, Showa University Fujigaoka Hospital, Yokohama, Kanagawa, Japan
[3]Department of Clinical Immuno Oncology, Clinical Research Institute for Clinical Pharmacology & Therapeutics, Showa University, Setagaya, Tokyo, Japan
[4]Department of Medicine, Division of Medical Oncology, Showa University Graduate School of Medicine, Shinagawa, Tokyo, Japan
[5]Pharmacological Research Center, Showa University, Shinagawa, Tokyo, Japan
[6]Department of Hospital Pharmaceutics, School of Pharmacy, Showa University Graduate School of Pharmacy, Shinagawa, Tokyo, Japan
[7]Department of Pharmacology, Showa University Graduate School of Dentistry, Shinagawa, Tokyo, Japan
[8]Department of Pharmacology, Showa University Graduate School of Pharmacy, Shinagawa, Tokyo, Japan
[9]Department of Gastroenterological Surgery, Tokai University School of Medicine, Isehara, Kanagawa, Japan
[10]Department of Toxicology, Showa University Graduate School of Pharmacy, Shinagawa, Tokyo, Japan
[11]Department of Physiology, Showa University Graduate School of Pharmacy, Shinagawa, Tokyo, Japan
[12]Department of Orthopaedic Surgery, Nihon University School of Medicine, Itabashi, Tokyo, Japan
[13]Department of Medicine, Division of Neurology, Showa University Graduate School of Medicine, Shinagawa, Tokyo, Japan
[14]Showa University Research Administration Center, Shinagawa, Tokyo, Japan

## AUTHOR ORCIDs

Yasunobu Amari  http://orcid.org/0009-0004-7645-8955
Masahiro Hosonuma  http://orcid.org/0000-0002-5861-7543

## FUNDING

| Funder | Grant(s) | Author(s) |
| --- | --- | --- |
| Showa University | | Yasunobu Amari |

## AUTHOR CONTRIBUTIONS

Yasunobu Amari, Conceptualization, Data curation, Formal analysis, Investigation, Writing – original draft | Takuya Mizukami, Formal analysis | Junya Isobe, Formal analysis | Yuki Azetsu, Formal analysis | Eiji Funayama, Formal analysis | Yuki Maruyama,

Formal analysis | Toshiaki Tsurui, Formal analysis | Kohei Tajima, Formal analysis | Aya Sasaki, Investigation, Methodology | Yoshitaka Yamazaki, Formal analysis | Ryota Nakano, Formal analysis | Yutaka Sano, Formal analysis | Atsushi Ishida, Formal analysis | Tatsuya Nakanishi, Formal analysis | Seiji Mochizuki, Formal analysis | Yuri Yoshizawa, Formal analysis | Sumito Kumagai, Formal analysis | Sakiko Yasuhara, Formal analysis | Kakei Ryu, Formal analysis | Tatsunori Oguchi, Formal analysis | Atsuo Kuramasu, Formal analysis | Kiyoshi Yoshimura, Formal analysis | Takehiko Sambe, Formal analysis | Sei Kobayashi, Formal analysis | Naoki Uchida, Formal analysis.

## DATA AVAILABILITY

All data pertaining to the study will be made available by the corresponding author upon reasonable request. The sequence data for the external auditory canal bacteria gene are available in the DDJB database (https://www.ddbj.nig.ac.jp/index.html) under accession numbers PRJDB18837.

## ETHICS APPROVAL

This study adhered to the principles outlined in the Declaration of Helsinki and was approved by the Ethical Review Committee of the Showa University School of Medicine (approval number: 22–272-B). All study participants provided informed consent.

## ADDITIONAL FILES

The following material is available online.

### Supplemental Material

**Supplemental material (Spectrum01698-24-s0001.pdf).** Fig. S1 to S4.

### Open Peer Review

**PEER REVIEW HISTORY (review-history.pdf).** An accounting of the reviewer comments and feedback.

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
