## [Reviewer comments · Microbiology Spectrum]

Microbiology Spectrum

Association between the *ABCC11* gene polymorphism-determined earwax properties and external auditory canal microbiota in healthy adults

Yasunobu Amari, Masahiro Hosonuma, Takuya Mizukami, Junya Isobe, Yuki Azetsu, Eiji Funayama, Yuki Maruyama, Toshiaki Tsurui, Kohei Tajima, Yoshitaka Yamazaki, Ryota Nakano, Yutaka Sano, Atsushi Ishida, Tatsuya Nakanishi, Seiji Mochizuki, Yuri Yoshizawa, Sumito Kumagai, Sakiko Yasuhara, Kakei Ryu, Tatsunori Oguchi, Atsuo Kuramasu, Kiyoshi Yoshimura, Takehiko Sambe, Sei Kobayashi, and Naoki Uchida

Corresponding Author(s): Masahiro Hosonuma, Showa Daigaku

Review Timeline:

Submission Date:	July 9, 2024
Editorial Decision:	July 24, 2024
Revision Received:	July 31, 2024
Editorial Decision:	September 13, 2024
Revision Received:	October 29, 2024
Accepted:	November 12, 2024

Editor: Jan Claesen

Reviewer(s): Disclosure of reviewer identity is with reference to reviewer comments included in decision letter(s). The following individuals involved in review of your submission have agreed to reveal their identity: Funmilola Abidemi Ayeni (Reviewer #2)

Transaction Report:

DOI: <https://doi.org/10.1128/spectrum.01698-24>

Re: Spectrum01698-24 (Association between the ABCC11 gene polymorphism-determined earwax properties and external auditory canal microbiota in healthy adults)

Dear Dr. Masahiro Hosonuma:

Thank you for the privilege of reviewing your work. Below you will find my comments, instructions from the Spectrum editorial office, and the reviewer comments.

Could you please submit a high resolution image for Figure 2? The one currently in the system does not have sufficient resolution to allow reading any of the text.

Revision Guidelines

Sincerely,
Jan Claesen
Editor
Microbiology Spectrum

Re: Spectrum01698-24R1 (Association between the ABCC11 gene polymorphism-determined earwax properties and external auditory canal microbiota in healthy adults)

Dear Dr. Masahiro Hosonuma:

Thank you for the privilege of reviewing your work. Below you will find my comments, instructions from the Spectrum editorial office, and the reviewer comments.

Thanks for submitting your paper to Spectrum. Your work has now been evaluated by two independent Reviewers who have raised some comments and suggestions (pasted below) to help you improve the manuscript. I would be happy to consider a revised version that addresses these comments in a point-by-point manner.

Revision Guidelines

Sincerely,
Jan Claesen
Editor
Microbiology Spectrum

Reviewer #1 (Public repository details (Required)):

The authors should submit the ABCC11 gene polymorphism raw sequencing data to a public database for free assess.

Reviewer #1 (Comments for the Author):

Comments and Suggestions for the Authors:

The manuscript by Amari et al., entitled "Association between the ABCC11 Gene Polymorphism Determining Earwax Properties and External Auditory Canal Microbiota in Healthy Adults," focuses on the ABCC11 gene polymorphism, which determines the type of earwax (wet or dry) and its relationship with specific commensal bacteria.

In the abstract section, the PICRUSt pathway results are not easily understandable: "PICRUSt analysis revealed significant enrichment of certain pathways, such as GLCMANNANAUT PWY, PWY 6107, PWY1G 0, and ENTBACSYN PWY in the GA group, whereas PWY0 41, PWY 7084, PWY 6731, VALDEG PWY, and PWY 722 were significantly enriched in the AA group (line 68)." Please clarify the meaning of the codes for each pathway.

The authors should add a brief explanation (in Tables and Figures) for each pathway identified in the PICRUSt analysis. This would significantly enhance the manuscript's readability. Additionally, in the results section, line 212, the authors should explain and name each of the pathways discussed.

Line 199: "The previously reported relationship between earwax characteristics and ABCC11 gene polymorphisms was consistent across all cases (Table 1)." The authors should explain how did they evaluated the earwax characteristics (wet or dry)?

In the discussion section, the content in lines 241 to 252 introduces themes more appropriate for the introduction. The authors should summarize these themes in the Discussion and introduce them in the Introduction section.

I suggest revising line 311 as follows: "In the present study, the ABCC11 311 gene polymorphisms CORRELATED WITH the ear canal bacterial microbiota. This indicated that alterations in the ear canal microbiota owing to ABCC11 gene polymorphisms may alter the microenvironment along with its metabolites thereby POTENTIALLY contributing to the development of otitis media pearls as a mechanism in the pathogenesis of cholesteatoma."

Sentences implying causality should be avoided throughout the manuscript, such as in line 347: "The ABCC11 gene polymorphism, which determines earwax properties, regulates the composition of the ear canal microbiota and its metabolic pathways." This sentence should be revised. For example: "The ABCC11 gene polymorphism, which determines earwax properties, may influence the composition of the ear canal microbiota and its metabolic pathways."

Line 341: The sentences are unclear: "This study assessed ABCC11 gene polymorphisms and ear canal microbiota in 21 healthy Japanese individuals. Therefore, certain external factors such as ethnic customs and climate were not assessed. Consequently, a comprehensive survey of other countries is required." The authors should rewrite this for clarity.

Minor comments:

In Figure 4b, the R^2 value is quite low. Nevertheless, the authors should clearly state in the legend that they conducted a linear regression and specify how the p-value was calculated.

In Figures 4c and 4d, the "None" group consists of only two subjects. The authors should include the statistical tests used in the figure legend.

Line 207: A comma is missing: "The results demonstrated the bacterial species composition of the ear canal in each sample and group and revealed various compositional proportions of certain bacterial species between the GA and AA groups (Figure 2a, b)."

The authors should submit the ABCC11 gene polymorphism raw sequencing data to a public database for free access.

Reviewer #2 (Public repository details (Required)):

16S sequence data

Reviewer #2 (Comments for the Author):

The study presents an interesting perspective on genomic, microbiome and physical characteristics. However, many important information is not written into the manuscript and the manuscript needs a major organization and arrangement of thoughts into a sequential and logical flow

Abstract - it started with the aim being genomic association with microbiome, it concluded with physical factors, such as sex,

age, and ear-cleaning habits affecting the microbiome. There should be a correlation between aim and results

Introduction

Results are being discussed in the last paragraph of introduction. Recast

Method

Ethical approval should be at the beginning of method section.

No information about the number, characteristics, date of recruitment, location etc of the participants

No stated method for PCR used for microbiota experiment.

How were the fungi population determined? The authors stated they used 16S sequencing, how was that used to get fungal population?

How were the ear cleaning / non cleaning population defined?

Discussion

Logical flow of discussion should be used.

There are contradictions in results being discussed and inference, for example, line 333 ``The ear-cleaning group exhibited fewer *Malassezia* and fungal species than the non-ear-cleaning group. This indicates that frequent removal of earwax may create an unfavorable microenvironment for fungal growth``. What they wrote here is that people who clean their ears have fewer fungal which is contradictory to the inference of having more fungal growth.

Comments and Suggestions for the Authors:

The manuscript by Amari *et al.*, entitled "Association between the ABCC11 Gene Polymorphism Determining Earwax Properties and External Auditory Canal Microbiota in Healthy Adults," focuses on the ABCC11 gene polymorphism, which determines the type of earwax (wet or dry) and its relationship with specific commensal bacteria.

In the abstract section, the PICRUSt pathway results are not easily understandable: "PICRUSt analysis revealed significant enrichment of certain pathways, such as GLCMANNANAUT PWY, PWY 6107, PWY1G 0, and ENTBACSYN PWY in the GA group, whereas PWY0 41, PWY 7084, PWY 6731, VALDEG PWY, and PWY 722 were significantly enriched in the AA group (line 68)." Please clarify the meaning of the codes for each pathway.

The authors should add a brief explanation (in Tables and Figures) for each pathway identified in the PICRUSt analysis. This would significantly enhance the manuscript's readability. Additionally, in the results section, line 212, the authors should explain and name each of the pathways discussed.

Line 199: "The previously reported relationship between earwax characteristics and ABCC11 gene polymorphisms was consistent across all cases (Table 1)." The authors should explain how did they evaluated the earwax characteristics (wet or dry)?

In the discussion section, the content in lines 241 to 252 introduces themes more appropriate for the introduction. The authors should summarize these themes in the Discussion and introduce them in the Introduction section.

I suggest revising line 311 as follows: "In the present study, the ABCC11 311 gene polymorphisms CORRELATED WITH the ear canal bacterial microbiota. This indicated that alterations in the ear canal microbiota owing to ABCC11 gene polymorphisms may alter the microenvironment along with its metabolites thereby POTENTIALLY contributing to the development of otitis media pearls as a mechanism in the pathogenesis of cholesteatoma."

Sentences implying causality should be avoided throughout the manuscript, such as in line 347: "The ABCC11 gene polymorphism, which determines earwax properties, regulates the composition of the ear canal microbiota and its metabolic pathways." This sentence should be revised. For example: "The ABCC11 gene polymorphism, which determines earwax properties, may influence the composition of the ear canal microbiota and its metabolic pathways."

Line 341: The sentences are unclear: "This study assessed ABCC11 gene polymorphisms and ear canal microbiota in 21 healthy Japanese individuals. Therefore, certain external factors such as ethnic customs and climate were not assessed. Consequently, a comprehensive survey of other countries is required." The authors should rewrite this for clarity.

Minor comments:

In Figure 4b, the R^2 value is quite low. Nevertheless, the authors should clearly state in the legend that they conducted a linear regression and specify how the p-value was calculated.

In Figures 4c and 4d, the "None" group consists of only two subjects. The authors should include the statistical tests used in the figure legend.

Line 207: A comma is missing: "The results demonstrated the bacterial species composition of the ear canal in each sample and group and revealed various compositional proportions of certain bacterial species between the GA and AA groups (Figure 2a, b)."

The authors should submit the ABCC11 gene polymorphism raw sequencing data to a public database for free access.

Oct 29, 2024

RE: Spectrum01698-24R1

Dear Dr. Jan Claesen and Editors of *Microbiology Spectrum*,

We are grateful to Editorial Board Member and two reviewers for their comments and invaluable suggestions. We have carefully revised the manuscript accordingly. We believe that the revised manuscript, with the changes and new data added, has taken into consideration essentially all of the comments, and hope that will have been improved to the satisfaction of the editors and reviewers. Please find below our point-by-point response to each of the comments made by the reviewers.

Reviewer comments:

Reviewer #1 (Public repository details (Required)):

The authors should submit the ABCC11 gene polymorphism raw sequencing data to a public database for free assess.

As reviewer pointed out, sequence data was insufficiently disclosed. The 16s sequence data was deposited to the DDJB database (accession numbers SSUB030880), and we have added the data to Materials and Methods section in the revised version of the manuscript (page 9; lines 186-188). On the other hand, ABCC11 gene polymorphism was analyzed by PCR analysis used specified primers, so no sequence data exist.

Reviewer #1 (Comments for the Author):

The manuscript by Amari et al., entitled "Association between the ABCC11 Gene Polymorphism Determining Earwax Properties and External Auditory Canal Microbiota in Healthy Adults," focuses on the ABCC11 gene polymorphism, which determines the type of earwax (wet or dry) and its relationship with specific commensal bacteria.

In the abstract section, the PICRUSt pathway results are not easily understandable: "PICRUSt analysis revealed significant enrichment of certain pathways, such as GLCMANNANAUT PWY, PWY 6107, PWY1G 0, and ENTBACSYN PWY in the

GA group, whereas PWY0 41, PWY 7084, PWY 6731, VALDEG PWY, and PWY 722 were significantly enriched in the AA group (line 68)." Please clarify the meaning of the codes for each pathway.

According to the reviewer's good suggestion, we have changed from pathway code number to understandable pathway name in the revised version of the manuscript (page 4; lines 69-73).

The authors should add a brief explanation (in Tables and Figures) for each pathway identified in the PICRUST analysis. This would significantly enhance the manuscript's readability. Additionally, in the results section, line 212, the authors should explain and name each of the pathways discussed.

According to the reviewer's suggestion, we have changed from pathway code number to understandable pathway name in the revised version (Table 2, Figure S4). In addition, we have discussed several pathways and added them to the fourth paragraph of the discussion section in the revised version of the manuscript (page 14; lines 307-page 15; lines 325).

Line 199: "The previously reported relationship between earwax characteristics and ABCC11 gene polymorphisms was consistent across all cases (Table 1)." The authors should explain how did they evaluated the earwax characteristics (wet or dry)?

In the first report on the relationship between the ABCC11 polymorphism and earwax characteristics, earwax characteristics were defined based on self-declaration (*Nat Genet* 38:324–330.). Therefore, we have defined them in the same way. We have cited the article on this definition and added it to the revised manuscript (page 7; lines 146-147).

In the discussion section, the content in lines 241 to 252 introduces themes more appropriate for the introduction. The authors should summarize these themes in the Discussion and introduce them in the Introduction section.

According to the reviewer's good suggestion, we have changed the content in lines 241

to 252 from Discussion to Introduction, and modified the first paragraph of the Discussion using logical flow in the revised version of the manuscript. (page 6; lines 108-115; page 12; lines 252-254)

I suggest revising line 311 as follows: "In the present study, the ABCC11 311 gene polymorphisms CORRELATED WITH the ear canal bacterial microbiota. This indicated that alterations in the ear canal microbiota owing to ABCC11 gene polymorphisms may alter the microenvironment along with its metabolites thereby POTENTIALLY contributing to the development of otitis media pearls as a mechanism in the pathogenesis of cholesteatoma."

As requested, we have corrected in the revised version of the manuscript (page 12; lines 263-267).

Sentences implying causality should be avoided throughout the manuscript, such as in line 347: "The ABCC11 gene polymorphism, which determines earwax properties, regulates the composition of the ear canal microbiota and its metabolic pathways." This sentence should be revised. For example: "The ABCC11 gene polymorphism, which determines earwax properties, may influence the composition of the ear canal microbiota and its metabolic pathways."

As requested, we have corrected in the revised version of the manuscript (page 16; lines 348-349).

Line 341: The sentences are unclear: "This study assessed ABCC11 gene polymorphisms and ear canal microbiota in 21 healthy Japanese individuals. Therefore, certain external factors such as ethnic customs and climate were not assessed. Consequently, a comprehensive survey of other countries is required." The authors should rewrite this for clarity.

As reviewer pointed out, it was our mistake. It was an unnecessary sentence, so we deleted it.

Minor comments:

In Figure 4b, the R² value is quite low. Nevertheless, the authors should clearly state in the legend that they conducted a linear regression and specify how the p-value was calculated.

Thanks for pointing this out. We have analyzed using the simple linear regression model in JMP and there was a statistically significant difference. The regression equation is shown below.

$$\text{The proportion of } Streptococcus \text{ spp} = -0.247 + 0.0100 * \text{age}$$

The statistical analysis method was added to the revised version of the manuscript. (page 9; lines 194-196, Figure Legend 4b)

In Figures 4c and 4d, the "None" group consists of only two subjects. The authors should include the statistical tests used in the figure legend.

According to the reviewer's suggestion, we have added the statistical tests to the revised version of the manuscript. (page 9; lines 196-page 10; lines 198, Figure Legend 4c, Figure S3b)

Line 207: A comma is missing: "The results demonstrated the bacterial species composition of the ear canal in each sample and group and revealed various compositional proportions of certain bacterial species between the GA and AA groups (Figure 2a, b)."

Thanks for pointing this out. It was our mistake. We have added a comma. (page 10; lines 211-214)

The authors should submit the ABCC11 gene polymorphism raw sequencing data to a public database for free assess.

As described above, we deposited the 16s sequence data.

Reviewer #2 (Public repository details (Required)):

16S sequence data

As requested, the 16s sequence data was deposited to the DDJB database (accession numbers SSUB030880), and we have added the data to Materials and Methods section in the revised version of the manuscript (page 9; lines 186-188).

Reviewer #2 (Comments for the Author):

The study presents an interesting perspective on genomic, microbiome and physical characteristics. However, many important information is not written into the manuscript and the manuscript needs a major organization and arrangement of thoughts into a sequential and logical flow

Abstract - it started with the aim being genomic association with microbiome, it concluded with physical factors, such as sex, age, and ear-cleaning habits affecting the microbiome. There should be a correlation between aim and results

Thanks for pointing this out. Abstract has been revised focused on the genomic association with microbiome. (page 4; lines 76-page 5; line 79)

Introduction

Results are being discussed in the last paragraph of introduction. Recast

According to the reviewer's suggestion, results have been deleted from introduction.

Method

Ethical approval should be at the beginning of method section.

As requested, we have moved ethical approval to the beginning of method section in the revised version of the manuscript (page 7; lines 137).

No information about the number, characteristics, date of recruitment, location etc of the participants

Thanks for pointing this out. We have added to the method section in the revised version of the manuscript (page 7; lines 143).

No stated method for PCR used for microbiota experiment.

How were the fungi population determined? The authors stated they used 16S sequencing, how was that used to get fungal population?

We appreciate the opportunity to clarify this point. We analyzed only 16s sequencing on for microbiota experiment. The fungi population was identified because the rare fungi have 16S ribosomal RNA. *Malassezia* identified in this study is one of the fungi with 16S ribosomal RNA. <https://www.ncbi.nlm.nih.gov/nucleotide/JX096926.1>

Although there is a large amount of “uncultured fungus” with 16S ribosomal RNA in the BLAST search, “uncultured fungus” is insufficient information for the main result. Therefore, we have moved the contents of “uncultured fungus” to the supplemental figure (Figure S3). Additionally, we emphasized and stated that our study was not focused on the fungal flora to the third paragraph of the discussion section in the revised version of the manuscript (page 14; line 302-306).

How were the ear cleaning / non cleaning population defined?

Thanks for pointing this out. It is written in the Date collection section of the method (page 8; lines 152-154).

Discussion

Logical flow of discussion should be used.

There are contradictions in results being discussed and inference, for example, line 333 ``The ear-cleaning group exhibited fewer *Malassezia* and fungal species than the non-ear-cleaning group. This indicates that frequent removal of earwax may create

an unfavorable microenvironment for fungal growth`. What they wrote here is that people who clean their ears have fewer fungal which is contradictory to the inference of having more fungal growth.

Exactly as the reviewer's mentioned, we needed to revise our Discussion section. We have substantially rewritten our Discussion section with the following paragraph structure using logical flow.

1. Discussion with previous studies on the ear canal bacterial flora in the context of genome-microbiome interactions (page 11; lines 238-page 13; line 270).
2. Discussion of the bacteria observed only in the GA group (page 13; lines 271-page 14; line 295).
3. Discussion of the bacteria observed only in the AA group (page 14; lines 296-page 14; line 306).
4. Discussion of the ABCC11 gene polymorphism-dependent metabolic pathways (page 14; lines 307--page 15; lines 325).
5. Discussion of the association of background factors with the bacterial flora (page 15; lines 326-page 16; lines 345).

Again, thank you for your comments.

Masahiro Hosonuma, M.D., Ph.D.

Department of Pharmacology, Showa University Graduate School of Medicine

1-5-8 Hatanodai, Shinagawa, Tokyo 142-8555, Japan

Tel: +81-3-3784-8125; Fax: +81-3784-8176

E-mail: masa-hero@med.showa-u.ac.jp

Re: Spectrum01698-24R2 (Association between the ABCC11 gene polymorphism-determined earwax properties and external auditory canal microbiota in healthy adults)

Dear Dr. Masahiro Hosonuma:

Thanks for addressing the Reviewers' comments in the revised version of your manuscript. I would like to congratulate you on the acceptance of your paper for publication in Spectrum.

Your manuscript has been accepted, and I am forwarding it to the ASM production staff for publication. Your paper will first be checked to make sure all elements meet the technical requirements. ASM staff will contact you if anything needs to be revised before copyediting and production can begin. Otherwise, you will be notified when your proofs are ready to be viewed.

Sincerely,
Jan Claesen
Editor
Microbiology Spectrum